# The Study Demands-Resources Framework: An Empirical Introduction

**DOI:** 10.3390/ijerph17145183

**Published:** 2020-07-17

**Authors:** Tino Lesener, Leonard Santiago Pleiss, Burkhard Gusy, Christine Wolter

**Affiliations:** Division of Prevention and Psychosocial Health Research, Freie Universität Berlin, 14195 Berlin, Germany; l.pleiss@fu-berlin.de (L.S.P.); burkhard.gusy@fu-berlin.de (B.G.); christine.wolter@fu-berlin.de (C.W.)

**Keywords:** study demands-resources framework, student engagement, student burnout, mental health, university students

## Abstract

Based on the well-established job demands-resources (JD-R) framework, in our study we introduce the novel study demands-resources (SD-R) framework. The SD-R framework allows the study of salutogenic and pathogenic effects of university settings on students’ health and well-being. Using a large sample of university students (*n* = 5660), our aim was to translate and validate the JD-R’s essential assumptions within the university context, and thus establish the SD-R framework. We performed structural equation modelling to examine these essential assumptions. As assumed, we found that study demands—the “bad things” at university—predict student burnout (β = 0.50), whereas study resources—the “good things” at university—predict student engagement (β = 0.70) as well as burnout (β = −0.35). Also, in line with the SD-R’s assumptions, student burnout predicts life satisfaction negatively (β = −0.34), whereas student engagement predicts life satisfaction positively (β = 0.29). Hence, we were able to introduce the novel SD-R framework and validate its core assumptions. The SD-R framework serves as an excellent theoretical basis to examine both the salutogenic and pathogenic effects of the study context on students’ health and well-being. However, the framework needs further longitudinal and meta-analytical verification in accordance with the research on the JD-R framework.

## 1. Introduction

The job demands-resources (JD-R) [1,2] framework is currently one of the most applied frameworks in occupational health psychology for examining the relationships between occupational characteristics and occupational health and well-being. For instance, a quick search in Google Scholar (April 2020) revealed about 20,000 hits for “Job Demands-Resources Model”, against only 4200 for Karasek’s well-known “Job Demand-Control Model”. The JD-R framework claims that specific occupational characteristics lead to well-being, which in turn affects health and performance. Its crucial advantage over other occupational frameworks is that it allows for the study of both the salutogenic and pathogenic effects of occupational characteristics on occupational health and well-being. Because of its widespread use, the JD-R framework is well-established and validated even meta-analytically, as well as longitudinally, as an excellent theoretical framework to examine occupational well-being in a broad range of organizations and occupational fields [3].

There is also a considerable amount of research on students’ health and well-being. Concepts like student engagement and student burnout, its antecedents, and outcomes gain more and more interest [4,5,6,7,8,9,10]. Several studies have focused on either the positive consequences of studying [11,12,13,14] or negative consequences of studying [15,16,17]. However, to date there is no validated theoretical framework that considers study characteristics and their positive and negative effects on students’ health and well-being explicitly and exclusively within the university context. 

Since from a psychological perspective, studying may be considered as work [14], the JD-R framework might be of great interest within the university context. However, to date there is no clear theoretical translation of the JD-R framework into the university context. A search in Medline, PsychArticles, and PsycINFO for “study demands-resources framework” reveals not even one hit. Hence, our aim was to apply the JD-R’s essential assumptions within the university context, and thereby introduce the novel study demands-resources (SD-R) framework. With this SD-R framework we are able to examine both the salutogenic and pathogenic effects of studying on students’ health and well-being, and thus improve their academic performance in the long run [11,18,19].

### 1.1. The Job Demands-Resources Framework 

The job demands-resources (JD-R) framework is currently the most popular theoretical framework to investigate the relationship between employee well-being and its antecedents and outcomes [20]. The JD-R framework proposes that poor job design and excessive job demands lead to burnout and health problems in the long run [1], and job resources lead to higher work engagement and better performance [1]. Furthermore, job resources are believed to mitigate burnout [1]. 

Job demands—the “bad things” at work [21]—are associated with sustained physical or mental effort, and therefore with certain physiological and psychological costs, whereas the job resources stated—the “good things” at work [21]—are functional in achieving work goals, reducing job demands, or stimulating personal development [1]. Burnout is defined as a consequence of extended exposure to specific job demands like intense physical, affective, and cognitive strain [22], and includes exhaustion, cynicism, and reduced professional efficacy [23], whereas work engagement is defined as “a positive, fulfilling, work-related state of mind that is characterized by vigor, dedication, and absorption” [24].

Even though some jobs are very different than others, the JD-R framework can be generalized across all occupational settings [25]. The JD-R framework has been applied and validated in various occupational and organizational contexts. Several reviews and meta-analyses have examined specific aspects of the JD-R framework [26,27,28,29]. A recent meta-analytic review has validated its essential assumptions longitudinally [3], thereby providing evidence for their causality. Hence, we can assume that the JD-R framework serves as a good framework to examine occupational health and well-being in employees across various occupational settings. 

According to Schaufeli and Bakker [25], this generalizability across all jobs is based on a common link between all jobs: All jobs have specific job demands and provide specific job resources. However, the applicability of the framework is not necessarily based on the concrete settings’ occupational background, but rather on the existence of specific demands and specific resources within this setting. Therefore, the framework might also be applicable to non-occupational settings, if they pose specific demands and provide specific resources.

### 1.2. JD-R’s Application within the University Context: The Study Demands-Resources Framework

Ouweneel et al. [14] claim that from a psychological perspective, studying may also be considered as work. Similar to employees, students, are faced with various demands. They are expected to attend lectures and seminars, and to invest time in self-studying [30]. Furthermore, they are demanded to manage high workload, in some disciplines up to 50 hours per week [31]. Perceived and actual workload is frequently associated with workplace burnout [2]. Further study-related demands arise from the pressure of behaving like a competent professional [30].

Besides study-related demands, students are also faced with social and developmental demands: Studying is often associated with moving out of the parents’ home, partly to a different location. Only 20% of students still share their living space with their parents [32]. Therefore, many students are not only required to meet the academic standards of studying, but also to adapt to a new environment, sometimes far from their hometown, family, and friends. Additionally, students are faced with financial demands: 69% of students are involved in working activities during the semester, and 59% of these students are absolutely required to do so to fund their subsistence. A total of 33% of students do not have enough financial resources to cover their monthly expenses, and 41% of students expect an increase of their study duration due to working activities [32]. However, studying does not exclusively pose demands. It also provides specific resources. While introducing the JD-R framework, Bakker et al. [1] introduced various job resources in occupational settings. Several of these are also applicable in the university context. Appreciation, autonomy, supervisor support—or, more accurately, teacher support—may be considered as study resources, which are in line with the above-mentioned definition of resources as good aspects of the specific setting [21]. Hence, various other constructs like social support and developmental opportunities may also be considered as study resources.

The potential to transfer the JD-R framework into the university context is based not only on the existence of specific demands and resources, but also on other striking similarities of those contexts. As Gusy et al. [33] pointed out, studying and working share multiple other key characteristics. Like working, studying full-time requires a massive time investment [31]. Like employees, students are involved in structured, organized, and coercive activities like learning for tests or attending classes [30]. Students are also expected to apply certain competences in order to finish externally created sets of tasks in a given period of time [34,35]. Similar to occupational settings, these activities are goal-oriented and evaluated [30], and their externally assessed quality may have impact on one’s future career. However, in contrast to working, studying serves to finance the livelihood only in the long term. In addition, students’ teachers are, in contrast to employees’ leaders, not authorized to give instructions. 

Moreover, there is empirical evidence for the applicability of some assumptions of the JD-R framework to students. Mokgele and Rothmann [36] found initial evidence for JD-R’s health impairment and motivational process in a sample of first-year students. Their results indicate that high study demands and a lack of study resources are positively associated with burnout. Additionally, the authors found a positive association between available study resources and engagement. Robins et al. [37] got similar results. Various personal and study-related resources (i.e. supervisor support, peer support, mindfulness, and optimism) were associated with student engagement, whereas study demands (i.e. workload and professional self-doubt) were associated with exhaustion [37]. Furthermore, general health was negatively associated with study demands and positively associated with study engagement [37]. These results underline the transferability of the two processes into student populations. However, there has not been a holistic approach yet to transfer the JD-R-framework into university context, and to validate this novel SD-R framework in a large sample of university students.

Due to these initial results and the similarity between working and studying, we assume that the concepts and essential assumptions proposed by the JD-R framework may be applicable within the university context. In line with Demerouti et al. [2], study demands—the “bad things” at university—can be defined as those physical, social, or organizational aspects of studying that require sustained physical or mental effort, and are therefore associated with certain physiological and psychological costs. Study resources—the “good things” at university—can be defined as positively valued physical, psychological, social, or organizational aspects of studying that are functional in achieving study-related goals, reducing study demands, or stimulating personal growth and development. Schaufeli et al. [23] define student engagement—similar to work engagement—as “a positive, fulfilling state of mind, that is characterized by vigor, dedication, and absorption.” Student burnout “refers to feeling exhausted because of study demands, having a cynical and detached attitude toward one’s study, and feeling incompetent as a student” [23]. 

Within this SD-R framework, we also imply two causal, essentially independent processes: the health impairment process and the motivational process. High study demands increase the risk for student burnout and lead to negative outcomes, such as health complaints, whereas high study resources play especially a motivational role, stimulate student engagement, mitigate student burnout, and foster positive outcomes, such as academic performance or commitment (see Figure 1).

As we have mentioned, working and studying share common ground in several aspects. However, working and studying also differ in several other aspects—for example, studying is less important for earning students’ living, or lecturers (compared to supervisors) are not authorized to give formal instructions. Thus, the major advantage of this novel SD-R framework is that it is much more specific and focused solely on the university context. Furthermore, it provides specific definitions of study demands and resources as well as student burnout and engagement. 

### 1.3. Aims and Hypotheses 

The central aim of our study is to introduce the SD-R framework theoretically, and to validate its essential assumptions within a large sample of university students from various disciplines. We distinguish specific study-related aspects as either study demands or study resources that affect students’ health and well-being. 

In accordance with the JD-R framework and the above-introduced health impairment and motivational process, we formulated the following hypotheses for the SD-R framework: High study demands are positively associated with student burnout.High study resources are positively associated with student engagement.High study resources are negatively associated with student burnout.High student burnout is negatively associated with students’ health.High student engagement is positively associated with students’ health.

## 2. Materials and Methods 

### 2.1. Sample

Our study was part of a national health monitoring program at German universities. In total, we sent out 24,679 e-mail invitations to students’ e-mail address. Of these, 5660 students from 295 different universities and 210 different disciplines completed the questionnaire, resulting in a response rate of 22.9%. The most represented universities were Universität Hamburg (3.3%), Johannes Gutenberg-Universität Mainz (3.0%), Universität zu Köln (2.8%), Humboldt Universität zu Berlin (2.5%), and Friedrich-Schiller-Universität Jena (2.2 %). The represented disciplines were Mathematics and Natural Sciences (22.5%), Social Sciences and Psychology (21.1%), Linguistics and Cultural Sciences (18.8%), Engineering (15.7%), Law and Economics (13.2%), and Medical or Health Sciences (8.8%).

The mean age of the respondents was 26.9 years (SD = 5.2 years), the youngest participants were 18 years old, and the oldest one was 69 years old. The majority of participants were female (62.2%), while 30 participants identified neither as male nor as female (0.5%). The respondents were on average enrolled in the third year of studying (mean = 3.1 years; SD = 1.9). On average, the students worked about 29.4 hours (SD = 16.1) for their studies, and an additional 14.4 hours (SD = 10.8) for their (part-time) jobs. The average income was €918 (SD = 469), and 6.6% of the students had at least one child. 

### 2.2. Measures

#### 2.2.1. Study Demands

To assess study demands, we operationalized challenging demands and time pressure. Adapted from Bakker’s Job demands-resources Questionnaire [38], the challenging demands scale aims to assess how cognitively challenging academic studies are for students. Its four items specify these demands in terms of concentration, precision, mental effort, attention, and multi-tasking. All items—for example, “Do your studies require a high degree of concentration?”—are answered on a Likert scale ranging from “never” (1) to “always” (6). A higher mean score indicates a higher level of (perceived) challenging demands. The internal consistency in our study was α = 0.85. Time pressure was measured with a self-constructed scale that has been used in various health monitoring surveys at German universities [39]. Its three items identify study demands induced by a subjective scarcity of time—for instance, “I am lacking time to properly process study-related tasks.“ All items were answered on a Likert scale ranging from “never“ (1) to “always“ (6). The internal consistency in our study was α = 0.73.

#### 2.2.2. Study Resources

To assess study resources, we operationalized student support, teacher support, and developmental opportunities. The constructs were measured with a self-constructed scale that has also been used in various health monitoring surveys at German universities [39]. Response categories for all items ranged from “never” (1) to “always” (6). A higher mean score indicates a higher level of (perceived) study resources. The three items to capture student support assess whether the individual can rely on fellow students for valid information and constructive feedback—for instance, “I am getting constructive feedback from my fellow students regarding my study-related performance.” The internal consistency in our study was α = 0.82. The three items to capture teacher support assess whether students can rely on their teachers for support regarding study-related issues and content-related questions—for instance, “I am getting advice from lecturers regarding study-related issues.” The internal consistency in our study was α = 0.85. The three items to capture developmental opportunities assess to what extent the individual believes his or her study program enhances career chances and imparts relevant knowledge and skills. One such item is “While studying, I am acquiring key qualifications that will be relevant to my future occupation. (e.g., communication, leadership, or social skills)”. The internal consistency in our study was α = 0.68.

#### 2.2.3. Student Burnout

To capture student burnout, we used the exhaustion sub-scale of the Maslach Burnout Inventory—Student Form (MBI-9-SF). This instrument, adopted for students, was introduced by Schaufeli et al. [23] and translated into German, then shortened and evaluated by Wörfel et al. [5]. The sub-scale consists of three items, such as “I feel tired when I get up in the morning and I have to face another day at the university.” The frequency of these experiences is scored from “never” (0) to “daily” (6). The sub-scale’s mean score is computed, and high scores are indicative of higher student burnout. The factorial validity of the abbreviated MBI-SF scales was successfully confirmed [5], and the internal consistency of the sub-scale in our study was α = 0.85

#### 2.2.4. Student Engagement

We used the ultra-short version of the Utrecht Work Engagement Scale—Student Form (UWES-3-SF) [4] with three items to capture student engagement. All items, for example “When I’m doing my work as a student, I feel bursting with energy”, are answered on a Likert scale ranging from “never” (0) to “always” (6). A higher mean score indicates a higher level of student engagement. The internal consistency of the scale was α = 0.85. 

#### 2.2.5. Health

To assess students’ health, we used the Satisfaction with Life Scale (SWLS) developed by Diener, Emmons, Larsen, and Griffin [40], which aims to quantify the subjects’ affective and cognitive judgement of their overall well-being. The SWLS is meant to provide a global assessment without inquiring about specific domains. The SWLS contains five items (such as “If I could live my life over, I would change almost nothing”), to which participants can respond using a seven-point Likert scale ranging from “strongly disagree” (1) to “strongly agree” (7). The sum of the ratings comprises the overall score ranging from 5 to 35, with higher scores indicating higher levels of satisfaction. The internal consistency in our study was α = 0.89.

### 2.3. Data Analysis

To test our hypotheses, we performed structural equation modelling (SEM) using Mplus version 8.2. Due to the non-normality of some study variables, and a sufficiently large sample size of *n* ≥ 400 [41], we used robust maximum likelihood estimation (MLR) for all model estimations. As recommended by Hu and Bentler [42], we assessed models’ goodness of fit by chi-square test statistic, root mean square error of approximation (RMSEA), standardized root mean square residual (SRMR), Tucker–Lewis index (TLI), and comparative fit index (CFI). A non-significant chi-square indicates good model fit. Since chi-square is sensitive to sample size, fit indices less sensitive to the number of observations were evaluated, too. An RMSEA value of less than 0.06 and a SRMR value of 0.08 or lower indicate good model fit [43]. For TLI and CFI, values of 0.90 may be interpreted as an acceptable fit [44]. 

## 3. Results

Means, standard deviations, and correlations of the study variables are reported in Table 1. We tested our hypotheses based on the SEM depicted in Figure 2.

### 3.1. Measurement Models 

We specified and tested the measurement model of the latent constructs prior to model testing. Challenging demands and time pressure served as indicators for study demands, and student support, teacher support, and developmental opportunities served as indicators for study resources, also based on the manifest variables depicted in Table 2. The overall measurement model showed an acceptable fit (χ^2^ (309) = 3811.61, *p* < 0.01; RMSEA = 0.05; SRMR = 0.06; TLI = 0.94; CFI = 0.95). All items loaded solidly on their respective factors (0.57 ≤ β ≤ 0.90; *p* < 0.001).

### 3.2. Structural Equation Model 

Our structural model to test our hypotheses is presented in Figure 3. The model showed an acceptable fit to the data: χ^2^ (313) = 4234.14, *p* < 0.01; RMSEA = 0.05; SRMR = 0.07; TLI = 0.93; CFI = 0.94. According to hypothesis 1, study demands strongly predicted student burnout (β = 0.50; *p* < 0.001), with time pressure yielding the highest predictive value (β = 0.92). In line with hypotheses 2 and 3, job resources were positively related to student engagement (β = 0.70; *p* < 0.001) and negatively related to student burnout (β = –0.35; *p* < 0.001). Developmental opportunities served as the strongest predictor (β = 0.69) for the latent construct of job resources. Finally, in line with hypotheses 4 and 5, both student burnout (β = –0.34; *p* < 0.001) and student engagement (β = 0.29; *p* < 0.001), predicted students’ health (i.e., their life satisfaction). Thus, as stated by the SD-R framework, all of our hypotheses were supported by our structural model. 

## 4. Discussion

The aim of our study was to apply the JD-R’s essential assumptions within the university context, and thereby introduce the novel SD-R framework. We defined study demands (the “bad things” at university) and study resources (the “good things” at university), and examined their relationships with students’ health and well-being. We were able to validate the health impairment and motivational process proposed by the SD-R framework. In line with hypotheses 1 and 4, and thus confirming the health impairment process, students’ study demands were positively associated with student burnout, which in turn was negatively associated with students’ life satisfaction. These results correspond to those of other studies on burnout in both occupational and university contexts [3,33,45,46]. Several demands, such as time pressure or workload [20,33,45], and outcomes like life satisfaction or depression [20,33,46] have been studied in both contexts. However, some demands (e.g., academic demands or worries about future competence) and outcomes (e.g., academic performance) have been studied exclusively within the university context [19,23,46], which is a strong argument for a context-specific SD-R framework. 

In line with hypotheses 2 and 5, and thus confirming the motivational process, students’ study resources were positively associated with student engagement, which in turn was positively associated with students’ life satisfaction. These results correspond with those of other studies on engagement in both occupational and university contexts [3,33,47]. As for demands, several resources have already been studied in both contexts. Examples of these are social support, role clarity, developmental opportunities, or autonomy [20,33,45,47]. However, there are also specific resources that have been studied solely in the university context as academic facilitators (e.g., services, availability of scholarships), choice of courses, or institutions for career preparation [48,49]. Furthermore, in line with hypothesis 3, study resources were negatively associated with student burnout. 

Since we know that almost 25% of students suffer from severe symptoms of exhaustion and anxiety, while almost half of students hold high levels of student engagement [50], we need a tool to assess the causes of these states of well-being. The SD-R framework provides a theoretical basis to examine the effects of the study context on students’ health and well-being. Both the motivational as well as the health impairment process proposed by the SD-R framework imply causality: study demands and study resources lead to student well-being, which in turn increases students’ health over time. With our cross-sectional data, we were not able to validate these causal assumptions properly. However, since the JD-R framework is well-established, and longitudinally as well as even meta-analytically validated [3], we can assume that the SD-R’s essential assumptions will hold in the long run, too. 

### Limitations

Despite our best efforts, the present study is not free of shortcomings.

First, even though we used a large sample from various German universities, our sample is not representative for German students in general. Female students, and students from social sciences were slightly overrepresented, whereas students from law and economics were slightly underrepresented. However, to our knowledge, this was the first study which gathered nationwide data on students’ health and well-being. With this large sample, we were able to get detailed insights into study-related aspects that affect students’ health and well-being.

Second, the use of cross-sectional data does not allow the causal inferences proposed by the SD-R framework. We firmly need longitudinal research to prove and validate the essential assumptions made by the SD-R framework properly. However, the JD-R framework, which is the theoretical and empirical basis for the novel SD-R framework, has been validated meta-analytically and longitudinally [3]. Hence, we are strongly confident that the SD-R’s assumptions should also be valid in the long run. Nevertheless, this needs to be examined in further research on the SD-R framework. 

Third, our study relies solely on self-reported data. A common method bias [51], and thus overestimation of observed effects, cannot be ruled out. Although we used reliable and valid scales, we strongly advise objective measures for either study demands and resources or students’ health and well-being in further studies. Furthermore, we used life satisfaction as the indicator for health and performance. Future research should also use other long-term outcomes (e.g., absenteeism as an indicator for students’ health, or marks or successful graduation as indicators for students’ performance). 

Fourth, we had to operationalize a selection of study demands and resources. We are firmly convinced that this selection is of central importance within the university context. However, there are many other relevant study demands and resources that should be considered in future research (e.g., autonomy, emotional demands). Due to its flexibility, the SD-R framework is open to a broad range of variables that can be operationalized as either study demands or study resources. In line with this, we want to acknowledge, that the reliability coefficient of one of our scales, namely developmental opportunities (α = 0.68), should be re-examined. However, since the scale consisted of three items only, the reliability coefficient can still be seen as acceptable [52]. We suggest that future research on the SD-R take advantage of its flexibility and examine a broad range of study demands and resources with valid and reliable scales.

Fifth, several authors suggest that within occupational settings, specific job demands may have a specific impact on work engagement [27,53]. Challenging job demands may have the potential to foster work engagement, whereas hindering job demands may diminish work engagement [27]. Unfortunately, since we included both challenging and hindering study demands, we were not able to hypothesize these specific effects within the university context. However, we strongly advocate for future research to examine the specific effects of (challenging and hindering) study demands on student engagement.

## 5. Conclusions

In our study, we were able to examine and validate the novel SD-R framework in a large sample of university students. The crucial advantage over other theoretical frameworks is that the SD-R framework allows for studying both salutogenic and pathogenic effects of the study context on students’ health and well-being. Hence, it serves as an excellent theoretical basis to assess both positive and negative consequences of studying within the university context. Universities are well-advised to limit study demands and strengthen study resources to ensure their students’ health and productivity.

## Figures and Tables

**Figure 1 ijerph-17-05183-f001:**
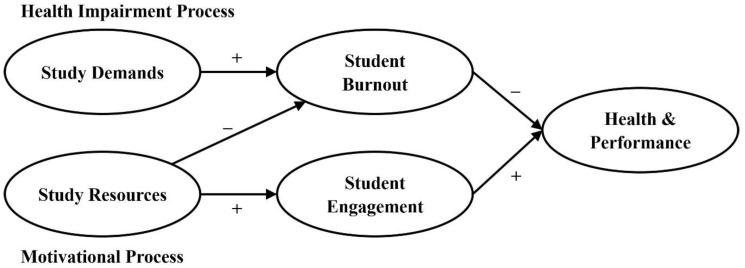
The study demands-resources (SD-R) framework.

**Figure 2 ijerph-17-05183-f002:**
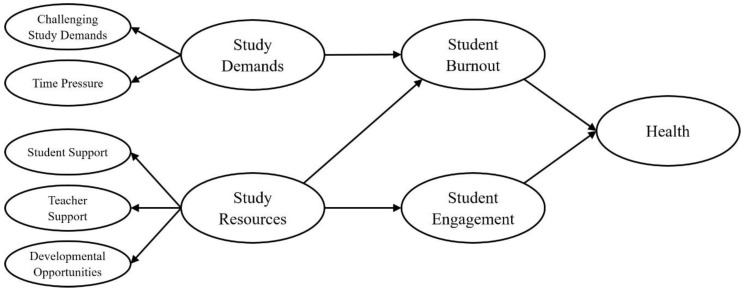
Hypothesized model. All exogenous latent constructs are represented by manifest variables shown in Table 2.

**Figure 3 ijerph-17-05183-f003:**
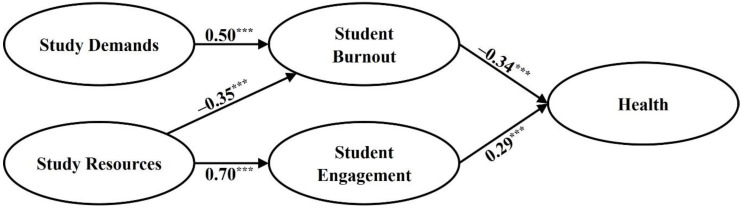
The final structural model. The manifest variables are not shown in this figure. *** *p* < 0.001.

**Table 1 ijerph-17-05183-t001:** Means, standard deviations, correlations, and reliability estimates (*n* = 5660).

	Variable	Mean	SD	1	2	3	4	5	6	7	8	9	10	11
1	Gender	–	–	–										
2	Age	26.9	5.2	0.08	–									
3	Income	918.38	468.97	0.04	0.29	–								
4	Challenging Demands	4.42	0.91	0.03	0.00	0.02	(0.85)							
5	Time Pressure	3.49	1.03	−0.06	0.10	−0.01	0.25	(0.73)						
6	Student Support	4.08	1.20	−0.07	−0.17	0.01	0.07	−0.24	(0.82)					
7	Teacher Support	3.81	1.14	0.03	−0.02	0.04	0.04	−0.23	0.38	(0.85)				
8	Developmental Opportunities	3.32	1.08	−0.02	−0.07	0.04	0.06	−0.17	0.37	0.38	(0.68)			
9	Student Burnout	2.15	1.51	−0.06	−0.06	−0.09	0.23	0.50	−0.25	−0.26	−0.21	(0.85)		
10	Student Engagement	3.34	1.19	0.03	0.05	0.07	0.18	−0.28	0.31	0.38	0.33	−0.42	(0.85)	
11	Life Satisfaction	24.60	6.50	−0.10	−0.15	0.09	0.00	−0.30	0.36	0.23	0.26	−0.40	0.38	(0.89)

Note: SD: standard deviation. Reliability estimates appear within parentheses.

**Table 2 ijerph-17-05183-t002:** Correlations of the study items.

		1	2	3	4	5	6	7	8	9	10	11	12	13	14	15	16	17	18	19	20	21	22	23	24	25	26
1	CD1																										
2	CD2	0.64																									
3	CD3	0.65	0.54																								
4	CD4	0.61	0.54	0.59																							
5	TP1	0.17	0.11	0.12	0.21																						
6	TP2	0.17	0.12	0.14	0.16	0.32																					
7	TP3	0.23	0.15	0.17	0.24	0.61	0.49																				
8	SS1	0.03	0.02	0.03	0.05	−0.15	−0.19	−0.17																			
9	SS2	0.06	0.05	0.06	0.06	−0.17	−0.21	−0.19	0.72																		
10	SS3	0.06	0.07	0.07	0.07	−0.16	−0.16	−0.15	0.51	0.59																	
11	TS1	0.03	0.03	0.07	0.02	−0.22	−0.08	−0.16	0.26	0.29	0.27																
12	TS2	0.02	0.04	0.07	0.00	−0.25	−0.11	−0.19	0.28	0.30	0.33	0.63															
13	TS3	−0.01	0.04	0.04	0.01	−0.23	−0.08	−0.17	0.24	0.28	0.32	0.64	0.71														
14	DO1	0.06	0.08	0.10	0.06	−0.17	−0.12	−0.14	0.25	0.28	0.30	0.33	0.36	0.36													
15	DO2	0.04	0.05	0.03	0.05	−0.11	−0.08	−0.09	0.23	0.24	0.23	0.24	0.25	0.24	0.39												
16	DO3	−0.01	0.02	−0.03	0.04	−0.09	−0.11	−0.09	0.25	0.25	0.23	0.17	0.21	0.23	0.37	0.50											
17	SB1	0.28	0.21	0.22	0.33	0.38	0.37	0.44	−0.13	−0.16	−0.12	−0.16	−0.19	−0.18	−0.17	−0.12	−0.11										
18	SB2	0.15	0.10	0.10	0.19	0.30	0.34	0.36	−0.17	−0.19	−0.16	−0.17	−0.21	−0.18	−0.19	−0.11	−0.10	0.66									
19	SB3	0.14	0.09	0.09	0.16	0.29	0.36	0.37	−0.26	−0.29	−0.22	−0.21	−0.25	−0.22	−0.22	−0.16	−0.15	0.63	0.68								
20	SE1	0.10	0.13	0.12	0.09	−0.25	−0.29	−0.26	0.24	0.29	0.28	0.26	0.32	0.29	0.33	0.21	0.21	−0.34	−0.40	−0.43							
21	SE2	0.14	0.16	0.20	0.13	−0.19	−0.16	−0.16	0.19	0.27	0.27	0.30	0.34	0.32	0.36	0.22	0.21	−0.26	−0.30	−0.33	0.66						
22	SE3	0.12	0.14	0.14	0.10	−0.17	−0.17	−0.15	0.16	0.22	0.22	0.24	0.30	0.26	0.28	0.16	0.15	−0.26	−0.31	−0.35	0.66	0.66					
23	LS1	−0.02	0.00	0.01	−0.03	−0.19	−0.27	−0.23	0.28	0.31	0.24	0.17	0.20	0.18	0.22	0.17	0.19	−0.30	−0.32	−0.35	0.36	0.30	0.28				
24	LS2	−0.03	−0.03	0.00	−0.04	−0.21	−0.21	−0.22	0.24	0.26	0.17	0.17	0.17	0.15	0.16	0.14	0.17	−0.27	−0.27	−0.31	0.26	0.21	0.19	0.65			
25	LS3	−0.01	−0.01	0.02	−0.04	−0.15	−0.25	−0.20	0.28	0.31	0.23	0.17	0.19	0.17	0.21	0.15	0.18	−0.31	−0.33	−0.37	0.35	0.29	0.27	0.76	0.65		
26	LS4	0.02	0.03	0.03	0.01	−0.14	−0.27	−0.18	0.28	0.30	0.23	0.16	0.19	0.16	0.22	0.16	0.18	−0.23	−0.26	−0.32	0.34	0.29	0.26	0.71	0.55	0.67	
27	LS5	0.00	0.00	0.03	0.00	−0.12	−0.22	−0.17	0.25	0.26	0.22	0.15	0.17	0.15	0.18	0.13	0.14	−0.24	−0.24	−0.29	0.29	0.25	0.23	0.61	0.47	0.61	0.58

Note: CD = challenging demands; TP = time pressure; SS = student support; TS = teacher support; DO = developmental opportunities; SB = student burnout; SE = student engagement; LS = life satisfaction.

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
