# Peer review of "The Study Demands-Resources Framework: An Empirical Introduction"

_ijerph, 2020, doi:10.3390/ijerph17145183_

Round 1

Reviewer 1 Report

Section Introduction: I suggest to shorten the text especially description of the JD-R Framework. 

Section Discussion: This section should be significantly extended. In the present form is rather presentation of the results. It would be worth presenting here other methodologies and research results on student well-being and compare them with yours. I have found two articles could be useful: 

Larcombe, Wendy and Tumbaga, Letty and Malkin, Ian and Nicholson, Penelope and Tokatlidis, Orania, Does an Improved Experience of Law School Protect Students Against Depression, Anxiety and Stress? An Empirical Study of Wellbeing and the Law School Experience of LLB and JD Students (September 16, 2012). Sydney Law Review, Vol. 35, No. 2, 2013; U of Melbourne Legal Studies Research Paper No. 603. Available at SSRN: https://ssrn.com/abstract=2147547 or http://dx.doi.org/10.2139/ssrn.2147547

Bodys-Cupak, I., Majda, A., Grochowska, A., Zalewska-Puchała, J., Kamińska, A., & Kuzera, G. (2019). Patient-related stressors and coping strategies in baccalaureate nursing students during clinical practice. Medical Studies/Studia Medyczne, 35(1), 41-47. https://doi.org/10.5114/ms.2019.84050

Author Response

Reviewer 1.

  1. Section Introduction: I suggest to shorten the text especially description of the JD-R Framework.

Answer: We agree that the introduction was too extensive. Hence, we have shortened and focused the introduction, especially the part of the JD-R framework (see p. 2).

  1. Section Discussion: This section should be significantly extended. In the present form is rather presentation of the results. It would be worth presenting here other methodologies and research results on student well-being and compare them with yours.

Answer: Thank you for this valuable advice for further improvement and the interesting articles you mentioned. We have restructured and expanded our discussion, and compared our results with both, evidence from the workplace and the university context (see p. 9). This is also in line with the suggestions of reviewer 4.

Reviewer 2 Report

The paper is in every aspect of high quality. Nevertheless, one thing should be explained at least in the Discussion. Why the authors do not also hypothesize that high study demands are positively associated with student engagement? The SD-R framework is based on JD-R framework and in some more recent versions of JD-R framework this association is considered. It is known that to stay motivated and engaged, and to perform well one needs to be challenged by the demands to a certain level. As also ancient wisdom says: If you tighten the string too much, it will snap, and if you leave it too slack, it will not play. Discussing this aspect will enhance the excellence of the paper.

Author Response

Reviewer 2.

  1. The paper is in every aspect of high quality. Nevertheless, one thing should be explained at least in the Discussion. Why the authors do not also hypothesize that high study demands are positively associated with student engagement? The SD-R framework is based on JD-R framework and in some more recent versions of JD-R framework this association is considered. It is known that to stay motivated and engaged, and to perform well one needs to be challenged by the demands to a certain level. As also ancient wisdom says: If you tighten the string too much, it will snap, and if you leave it too slack, it will not play. Discussing this aspect will enhance the excellence of the paper.

Answer: Thank you very much for your positive feedback and your valuable advice. We totally agree with you that – in line with your argumentation – it would be a substantial improvement to integrate the discussion of challenging and hindering demands in our manuscript. However, since we did not differentiate between challenging and hindering demands in this study, we were not able, we were not able formulate specific hypotheses.  Our intention was to first introduce a basic version of the SD-R framework which has never been done before. Since the basic assumptions of the SD-R framework have been confirmed in our study, future studies might consider more elaborated versions of the framework differentiating also between challenging and hindering demands. However, we now mention this crucial aspect in our discussion (p. 10). We hope, that this aspect will be considered in further studies.

Reviewer 3 Report

The idea of a Study Demands-Resources Framework is attractive. The paper is well written, with good method and analysis. However, i have some suggestions for revision : 

Minor point. 

p2, l64-68. Burnout dimensions may also be shortly defined, as for engagement. 

p3, l122-134. I think there is more than first empirical evidence for the applicability of some assumptions of the JD-R framework to students. Many study have used JD-R in students (university and school). Also and for ex. Salmela-Aro and Upadyaya (2014) already showed the de Demands-resources model can be usefully applied to the school context, including the two processes leading to engagement and burnout. 

Major point

To me, its not clear what is the specifity of the Study Demands-Ressources model. The demands-resources model seems to be relevant in school and university contexte so is there a need for a framework extension ? I think the authors should explain (in introduction and discussion) more why this extension is needed and what its specificity is. In the limitations part, line 325 authors said that there are many other relevant study demands and resources that should be considered in future research. I think this could be something to discuss (e.g. review what demands and/or ressources are the same in the Job and Study model and what are the specific ones in the study model) 

Author Response

Reviewer 3.

Minor points

  1. p2, l64-68. Burnout dimensions may also be shortly defined, as for engagement.

Answer: Thank you for your attentive comment. We have added the sub-dimensions for burnout (see p. 2).

  1. p3, l122-134. I think there is more than first empirical evidence for the applicability of some assumptions of the JD-R framework to students. Many study have used JD-R in students (university and school). Also and for ex. Salmela-Aro and Upadyaya (2014) already showed the de Demands-resources model can be usefully applied to the school context, including the two processes leading to engagement and burnout.

Answer: We totally agree with you that there is more than »first« evidence for the applicability of the JD-R framework within university students. Hence, we reformulated this part of our introduction (see p. 3).

Major point

  1. To me, its not clear what is the specifity of the Study Demands-Ressources model. The demands-resources model seems to be relevant in school and university contexte so is there a need for a framework extension ? I think the authors should explain (in introduction and discussion) more why this extension is needed and what its specificity is. In the limitations part, line 325 authors said that there are many other relevant study demands and resources that should be considered in future research. I think this could be something to discuss (e.g. review what demands and/or ressources are the same in the Job and Study model and what are the specific ones in the study model)

Answer: Thank you very much for this valuable comment. We regret to not having clarified why we think it to be very important to have a specific framework that examines health and well-being of university students. In our view, studying and working share common ground in several aspects. However, since they also differ substantially in other aspects (e.g. specific study demands and resources that are not relevant in occupational contexts), we firmly believe that the introduction of this SD-R framework and especially of specific definitions of study demands and resources is necessary to conduct further research on students’ health and well-being. Hence, we have added this argumentation in our introduction (see p. 4). Further, we now discuss other relevant and specific study demands and resources in our discussion (see p. 9). We hope that our revised manuscript addresses your concerns sufficiently and we can now make even clearer the importance of this novel SD-R framework.

Reviewer 4 Report

The study is of great interest, I congratulate the authors on the sample size!
The introduction reviews the evidence on the Job Demands-Resources Model extensively. The methodology is complete.

The results correspond to the objectives set. However, it would have been interesting to know the profile of the students with the highest burnout and the profile of the students with the highest engagement. I assume these data are available. Also, I’d like to know the demographic factors associated with these theoretical constructs and with satisfaction with life.

The results are generic in a large sample of students. So, I find it interesting to highlight that the results correspond to the evidence that exists in the workplace, especially the inverse relationship between burnout and engagement.

Minor comments: In my opinion, the phrase on the Google Academic search  in the introduction, although it is an example, it should be completed with a scientific search or withdraw it.

Author Response

Reviewer 4.

  1. The results correspond to the objectives set. However, it would have been interesting to know the profile of the students with the highest burnout and the profile of the students with the highest engagement. I assume these data are available. Also, I’d like to know the demographic factors associated with these theoretical constructs and with satisfaction with life.

Answer: Thank you for this idea to improve our manuscript. We totally agree with you, that it would be very interesting to know these profiles of high burnout and high engagement students. However, our aim was to focus on the study-related demands and resources and its associations with student health and well-being. Furthermore, Salmela-Aro & Read (2017) have already identified profiles of student engagement and burnout in higher education. Hence, we decided to not inflate the manuscript with additional research questions. Concerning your second suggestion, we have added demographic factors in our correlation matrix (p. 7).

  1. The results are generic in a large sample of students. So, I find it interesting to highlight that the results correspond to the evidence that exists in the workplace, especially the inverse relationship between burnout and engagement.

Answer: Thank you for this valuable advice. We now highlight that our results are in line with other research in both occupational and university contexts (see p. 9).

  1. Minor comments: In my opinion, the phrase on the Google Academic search in the introduction, although it is an example, it should be completed with a scientific search or withdraw it.

Answer: Thank you for your feedback. We have reconducted our search in Medline, PsycARTICLES, and PsycINFO and reformulated that evidence (see p. 2).

Round 2

Reviewer 3 Report

Thank you to the authors for those modifications.